# Calcium Homeostasis in the Development of Resistant Breast Tumors

**DOI:** 10.3390/cancers15112872

**Published:** 2023-05-23

**Authors:** Desirée Martin-García, Teresa Téllez, Maximino Redondo, Marilina García-Aranda

**Affiliations:** 1Surgical Specialties, Biochemistry and Immunology Department, Faculty of Medicine, University of Málaga, 29010 Málaga, Spain; desirermg@uma.es (D.M.-G.); teresatellez@uma.es (T.T.); 2Instituto de Investigación Biomédica de Málaga-Plataforma BIONAND (IBIMA-BIONAND), Severo Ochoa, 35, 29590 Málaga, Spain; marilina.garcia.sspa@juntadeandalucia.es; 3Red de Investigación en Servicios de Salud en Enfermedades Crónicas (REDISSEC) and Red de Investigación en Cronicidad, Atención Primaria y Promoción de la Salud (RICAPPS), Instituto de Investigación Biomédica de Málaga (IBIMA), 29590 Málaga, Spain; 4Research and Innovation Unit, Hospital Costa del Sol, Autovia A-7 km 187, 29602 Marbella, Spain

**Keywords:** breast cancer, calcium, resistant tumors, targeted therapy, homeostasis, Ca^2+^ channels

## Abstract

**Simple Summary:**

Improving the response of breast cancer patients by designing and applying the most appropriate treatment for each case is a major scientific challenge. Given the role of intracellular calcium in cell proliferation, apoptosis evasion and cell resistance, in this review, we discuss its potential for the development of new pharmacological treatments to treat the disease.

**Abstract:**

Cancer is one of the main health problems worldwide. Only in 2020, this disease caused more than 19 million new cases and almost 10 million deaths, with breast cancer being the most diagnosed worldwide. Today, despite recent advances in breast cancer treatment, a significant percentage of patients will either not respond to therapy or will eventually experience lethal progressive disease. Recent studies highlighted the involvement of calcium in the proliferation or evasion of apoptosis in breast carcinoma cells. In this review, we provide an overview of intracellular calcium signaling and breast cancer biology. We also discuss the existing knowledge on how altered calcium homeostasis is implicated in breast cancer development, highlighting the potential utility of Ca^2+^ as a predictive and prognostic biomarker, as well as its potential for the development of new pharmacological treatments to treat the disease.

## 1. Introduction

### 1.1. Epidemiology and Risk Factors for Breast Cancer

Cancer is a major public health problem worldwide, as the World Health Organization (WHO) estimates that by 2040, there will be 28.9 million new cases that will cause more than 16.2 million deaths annually. Although breast cancer mainly affects women, it has become the most diagnosed malignancy in the general population, precisely because of the number of cases diagnosed in women, surpassing lung cancer, causing more than 2.2 million new cases (11.7%) and more than 680,000 deaths in 2020 alone, corresponding to one in six cancer deaths in women [1]. The burden of breast cancer is expected to increase year after year, despite having a high remission rate once the disease is identified and treated prior to progression to metastatic disease [2,3,4].

The risk factors associated with this disease include both intrinsic and extrinsic factors. The intrinsic factors are not avoidable and are associated with genetic and epigenetic characteristics [5], including mutations in autosomal dominant genes, such as breast cancer 1 (*BRCA1*) and breast cancer 2 (*BRCA2*) [6]; mutations in moderate-risk genes, such as the CHK2 serine/threonine protein kinase gene (*CHEK2*), the ataxia telangiectasia gene (*ATM*), and the partner and localize of BRCA2 gene (*PALB2*); or low-frequency variations, such as single-nucleotide polymorphisms (SNPs). The extrinsic factors are avoidable factors, such as sedentary lifestyles, obesity, alcohol, tobacco or drug use, use of birth control pills or hormone replacement therapies, and breast density [7,8]. In addition, it was observed that parity and age at menarche are implicated in the risk of breast cancer [9,10] and that different sociodemographic characteristics, such as lack of education, presence of anxiety or depression, or above-average comorbidities, cause a delay in the treatment of patients [11].

### 1.2. Heterogeneity of Breast Cancer: Progression of the Disease and Histological and Molecular Classifications

Generally, tumors develop following a sequence of initial lesions or alterations, hyperplasia, dysplasia, carcinoma in situ and invasive carcinoma. Traditionally, the histological classification scheme for breast cancer has been divided into (1) carcinoma in situ, which comprise noninvasive tumors with potentially malignant intraductal cells confined to the ducts (ductal carcinoma in situ) or lobules (lobular carcinoma in situ) from which cells can evolve uncontrollably to invasive, or (2) infiltrative carcinoma, in which neoplastic cells have penetrated stroma [12]. Although current consensus recognizes invasive ductal and lobular carcinomas, it was reported that most of these tumors arise in terminal ductal–lobular units (TDLUs) regardless of the histologic type [13]. Ductal carcinoma is the most commonly diagnosed invasive breast cancer, accounting for 50–75% of cases, followed by lobular carcinoma (5–15%) and mixed ductal/lobular carcinomas [14].

As for the progression of the disease, the traditional TNM staging system, which is based on tumor (T) anatomic features, regional lymph nodes (N) involvement, and the presence or absence of metastases (M) (Figure 1), has been the gold standard for determining patient prognosis over the last 70 years [15]. Although it was reported that breast cancers diagnosed at stages I and II have an overall survival of over 95%, up to 72% when diagnosed at stage III and reduced to 22% when diagnosed at stage IV [16], this anatomically based system is not enough to address the tumor biology and guide decision-making and treatment planning for all breast cancers, e.g., the triple-negative subtype is difficult to manage [17]. The eighth edition of the American Joint Committee on Cancer, announced in 2017 and globally adopted on 1 January 2018, also integrated biomarkers such as tumor grade, hormone receptor status, expression of the human epidermal growth factor receptor (EGFR) family member HER2 (Human Epidermal Growth Factor Receptor 2/ErbB2 receptor tyrosine kinase 2) or multigene panel status for certain sub-groups, resulting in different prognostic stages for tumors with virtually identical histologic types [18]. These have highlighted the important role of breast cancer heterogeneity in the correct clinical management of the disease.

With respect to hormone receptors (estrogen receptor (ER∝) and progesterone receptor (PR)) and human epidermal growth factor receptor 2 (HER2):Hormone-receptor-positive breast tumors, which account for 75% of breast carcinomas, are classified into luminal A breast tumors (50–60% of diagnosed cases)—which are ER-positive and/or PR-positive, HER2-negative and Ki67 < 14% [19] with low histological grade, and have a low mitosis proportion number and good prognosis—and luminal B tumors (15–20% of diagnosed cases)—which are defined as ER-positive and/or PR-positive (PR < 20% + Ki67 ≥ 14%), HER2-negative or ER-positive and/or PR positive/negative (any PR-positive and any Ki67) and HER2-positive. Luminal B tumors usually have a more aggressive phenotype, both by histologic grade and proliferative Ki67 index, and worse prognosis than luminal A tumors [20].HER2-enriched tumors, which account for approximately 15–20% of breast tumors, present HER2 overexpression [21]. These tumors do not express estrogen or progesterone receptors and are characterized by the overactivation of signaling pathways involved in increased cell proliferation (Ras/MAPK mitogen-activated protein kinases and PI3K/AKT phosphoinositide 4-kinase/protein kinase B), with increased risk of metastasis and a more aggressive phenotype than luminal tumors [22].Basal-like tumors are characterized by a lack of HER2 overexpression and the absence or low levels of ER/PR expression. Among basal-like tumors, the triple-negative subtype, which constitutes approximately 80% of basal-like tumors and 10–15% of breast carcinomas, is defined by the lack of hormone receptors (ER-, PR-), the lack of HER overexpression (HER2-) and being cytokeratin-5/6-positive (CK5/6+) and/or Epidermal-Growth-Factor-Receptor-positive (EGFR+) [23].

The molecular classification of breast cancer has allowed for the development of personalized therapeutic options, which have greatly improved patient response and prognosis. Since estrogen receptors are steroid hormone receptors that induce the production of growth factors, such as Epidermal Growth Factor (EGF), Insulin-like Growth Factor-1 (IGF) or Transforming growth factor alpha (TGFα), which stimulate tumor cell proliferation, competitive estrogen–estrogen receptor inhibitors have shown their utility to decrease tumor cell proliferation [24,25,26]. In such a manner, targeted treatments based on the use of monoclonal antibodies revolutionized the treatment for HER2-enriched breast tumors [27]. Unfortunately, although TNBCs are associated with poor long-term prognosis, higher probabilities of recurrence over time, and high probabilities of local and distant recurrence [28,29], no effective therapy has yet been approved for the targeted treatment of these tumors. The 5-year overall survival for non-metastatic disease is 85% for TNBC stage I patients compared with 94–99% for stage I patients with hormone-receptor-positive and HER2-positive breast tumors [25]. However, the overall 5-year survival rate for patients with metastatic disease is 22% [30].

These molecular classifications with major predictive and prognostic implications opened the way to histologic-independent personalized therapies, such as poly-ADP ribose polymerase (PARP) inhibitors for the treatment of tumors with mutations in *BRCA1* and *BRCA2* genes (present in up to 5% of breast cancer patients [31]) by preventing tumor cells with BRCA1/BRCA2 mutations from repairing DNA damage caused by cytotoxic chemotherapy [32].

As for more advanced transcriptomic analyses, i.e., assays that analyze the expression of multiple genes with the aim of providing prognostic and predictive information about breast cancer patients, these should be used at the time of initial diagnosis, not after relapse, and help to make therapeutic decisions when this is not clearly based on traditional clinicopathologic features [33,34]. Oncotype DX is a test developed by the Genomic Health, Inc., laboratory that analyzes 16 cancer-related genes for the diagnosis and prediction of ER-positive and HER2-negative cancer patients. MammaPrint analyzes the expression of 80 genes that allow the tumor to be categorized as luminal A, luminal B or basal-like [34]. Prosigna is a test designed for HER-positive postmenopausal patients and analyzes the expression of 50 genes to categorize the tumor into luminal A, luminal B, basal-like or HER2-enriched subtypes [33].

### 1.3. Conventional Treatments for Breast Cancer

The conventional treatments used to treat breast cancer are surgery, radiotherapy, chemotherapy, hormone therapy and immunotherapy, used alone or in combination.

Breast cancer surgery usually consists of two options: conservative surgery and mastectomy. Currently, breast-conserving surgery has replaced mastectomy, as the overall and disease-free survival rates are equivalent to this radical procedure. In addition, current early diagnosis programs have made the early detection of tumors possible, which allows for avoiding radical mastectomies in most cases. Although with conservative surgery, only the tumor mass is removed, sometimes it is necessary to remove more than 20% of the normal breast tissue surrounding the tumor, which has implications for the physical, emotional, and mental health of the patient. In recent years, the implementation of neoadjuvant chemotherapy has allowed for reducing the tumor size before surgery and, therefore, more conservative surgical interventions. Moreover, the incorporation of sentinel lymph node biopsy into surgery has made it possible to reduce the extent of surgery without compromising the prognostic value [35,36,37] since patients with one or two positive sentinel nodes should no longer undergo axillary lymphadenectomy [38].

Radiation therapy by means of X-rays or gamma rays is usually used to eliminate possible cancer cells that remain in the area after surgery. This additional component of breast-conserving therapy, which includes strong enough radiation doses that ensure the complete elimination of malignant cells [39], can be omitted in patients with limited life expectancy, adjuvant endocrine therapy, negative nodes, and hormone-receptor-positive or HER2-negative tumors [37].

Chemotherapy can be applied both before surgery to reduce tumor size (neoadjuvant chemotherapy) and avoid mastectomy and/or after surgery (adjuvant chemotherapy), while always considering the tumor size, hormone and HER2 receptor status, as well as the lymph node status [40]. Adjuvant chemotherapy is usually recommended for patients with a high risk of disease recurrence and usually involves combined treatment with taxanes and anthracyclines. For patients at low risk, anthracyclines are usually omitted [41].

Hormone therapy is the first-line option for all patients with ER-expressing breast cancer, where tamoxifen (Nolvadex, AstraZeneca Pharmaceutics, Cambridge, UK), which is a selective receptor estrogen modulator, is the drug commonly used because of its ability to reduce disease recurrence by half [42]. In postmenopausal women, tamoxifen is replaced by aromatase inhibitor drugs, which also target the estrogen signaling pathway, such as anastrozole (Arimidex, AstraZeneca Pharmaceutics, United Kingdom) or letrozole (Femara, Novartis Pharma, Basel, Switzerland) [43], as they produce a greater reduction in breast cancer recurrence than tamoxifen alone [44]. However, the best way to use these therapies is still uncertain [45].

The use of monoclonal antibodies opened a new era in the fight against breast with targeted treatments (Table 1). HER2, which plays a key role in tumor growth by activating different signaling pathways closely linked to cell proliferation, can be targeted with Trastuzumab (Herceptin, Roche Registration GmbH, Grenzach-Wyhlen, Germany) and pertuzumab (Perjeta, Roche Registration GmbH, Germany), which are two monoclonal antibodies that inhibit HER2 through the extracellular domain of the receptor [46], thereby blocking the signaling pathways it controls, and thus, exerting a considerable antitumor effect [47]. In 1998, Trastuzumab became the first monoclonal antibody approved by the FDA to treat HER2-positive breast cancer patients. Pertuzumab was approved in 2013 by the FDA for use in combination with Trastuzumab for HER2-positive patients at risk of relapse [48], which is a scheme that has shown good tolerability and a decrease in associated side effects [49]. In 2021, the FDA approved Margetuximab (Margenza, Macrogenics, Rockville, MD, USA) as a monoclonal antibody against HER2 for patients with HER2-positive metastatic breast cancer [50,51], the use of which in combination with chemotherapy significantly improves overall survival, although with important associated adverse effects [52].

Other HER2-associated monoclonal antibodies include epidermal growth factor receptor (EGFR) and transforming growth factor alpha (TFGα), whose binding activates the PTEN/I3K/Akt/mTOR and Ras/Raf/MEK intracellular signaling pathways. These are directly involved in cell proliferation and apoptosis. Inhibitors of kinases (TKIs) inhibitors at the extracellular domain level of HER2, such as lapatinib (Tyverb, Novartis Europharm Limited, Dublin, Ireland), neratinib (Nerlynx, Pierre Fabre Medicament, Paris, France), tucatinib (Tukysa, Seagen B.V., Schiphol, The Netherlands) and pyrotinib (AiRuiNi, Jiangsu Hengrui Pharmaceutical Group Co., Ltd., Lianyungang, China), are noteworthy in this regard [53]. Lapatinib is a HER2 and EGFR tyrosine kinase inhibitor at the kinase ATP binding site level and was approved in 2018 by the FDA for HER2-positive patients in combination with other anti-HER2 agents, such as trastuzumab [54]. Neratinib binds to the tyrosine kinase domain of HER2 and was approved by the FDA in 2018 likewise for HER2-positive stage I to III patients who received adjuvant therapy with trastuzumab [55]. Tucatinib, which is highly selective against HER2, was approved in 2020 by the FDA for advanced-stage HER2-positive patients and in combination with trastuzumab [56]. Finally, pyrotinib is a HER inhibitor that was approved in 2018 in China for advanced-stage HER2-positive patients who received prior chemotherapy [57].

The recent incorporation of drug–antibody conjugates against breast cancer represent an innovative therapeutic approach that combines the high specificity and antitumoral properties of monoclonal antibodies with the potent cytotoxic activity of small molecule drugs [58]. Examples of these conjugates are (1) trastuzumab emtansine (T-DM1) (Kadcyla, Roche Pharma AG, Germany), which includes trastuzumab, and a maitansinoid deriv-ative, which depolymerizes cell microtubules and triggers cell apoptosis [59], were ap-proved by the FDA in 2013 for patients with HER2-positive metastatic breast cancer [60], and (2) Trastuzumab deruxtecan (DS-8201a) (Enhertu, Daiichi Sankyo Europe GmbH, Munich, Germany), which was approved by the FDA in 2020 for the treatment of HER2-positive breast cancers and is composed of trastuzumab, a maleimide, and a topoisomerase inhibitor [58,61].

### 1.4. Targeted Therapies for Breast Cancer

Angiogenesis is a process directly involved in tumor development, as tumor formation depends on the formation of new blood vessels and influences the appearance of metastasis [62]. Although it is a process controlled by a variety of factors, vascular endothelial growth factor A (VEGF-A) is among those mainly responsible [40]. The human monoclonal antibody anti-VEGF-A Bevacizumab (Avastin, Roche Registration GmbH, Germany) is among the most prominent antiangiogenic drugs for angiogenesis inhibition [63,64]. Despite causing many side effects, such as bleeding, skin rashes and hypertension [65], Bevacizumab was approved in 2008 by FDA for the treatment of HER2-negative breast cancer in combination with paclitaxel (Taxol, Teva Pharma, Madrid, Spain) or capecitabine (Kern Pharma, Barcelona, Spain) [66].

Cyclin-dependent kinases (CDKs), such as the kinase cyclin D/cdk4/6, are key enzymes in cell progression, tumor development and clonal expansion [67]. CDK4/6 inhibitors, such as palbociclib (Ibrance, Pfizer, Ixelles, Belgium), ribociclib (Kisqali, Novartis Europharm Limited, Ireland) and abemaciclib (Verzenios, Eli Lilly Nederland B.V., Utrecht, The Netherlands), were approved in 2017 by the FDA for the treatment of HER2-positive or -negative breast tumors, in combination with endocrine therapy. Although this scheme can cause neutropenia as the main side effect, it is usually well tolerated and has led to a significant improvement in patient overall survival [68].

The PI3K/Akt/mTOR pathway plays a fundamental role in cell proliferation, survival and development [69], and is altered in breast cancer [70]; therefore, efforts have focused on trying to inhibit the various components that make up this signaling pathway. The PI3K inhibitor alpelisib (Piqray, Novartis Europharm Limited, Ireland) was the first FDA-approved breast cancer drug for hormone-receptor-positive and HER2-negative patients. Its approval in 2020 was under its combined use with fulvestrant (AstraZeneca, UK), which is an estrogen receptor antagonist [71], and its most common side effect is hyperglycemia [72]. For its part, everolimus (Afinitor, Novartis Europharm Limited, Ireland), which is an inhibitor of the mTORC1 complex, was approved by the FDA in 2009 for patients with hormone-receptor-positive, HER2-negative advanced breast cancer in combination with exemestane (Exemestane Sandoz, Sandoz Farmacéutica, Madrid, Spain), which is a steroid aromatase inhibitor. Like Alpelisib, Everolimus causes hyperglycemia as a major side effect, as both affect lipid metabolism [73].

Currently, for Akt kinase, the inhibitor ipatasertib (GDC-0068, RG7440) is still under development for the treatment of locally advanced/metastatic inoperable TNBC. In the preclinical phase, it demonstrated efficacy in inhibiting the PI3K/AKT pathway [74]. In phase Ib, the combination of this drug with paclitaxel (Taxol, Teva Pharma, Spain) evidenced good tolerance [75], while in phase II, it managed to improve tumor-progression-free survival [76,77]. However, recent phase III results show that adding this drug does not improve the efficacy of treatment with paclitaxel (Abraxane, Bristol-Myers Squibb Pharma, Dublin, Ireland) [78].

Mutations that cause errors in the DNA replication process, as well as those affecting the DNA repair machinery, are common in the development of cancer [79]. Poly ADP-ribose polymerase (PARP) enzymes, which are involved in DNA repair, and members of the BER pathway, which is the base excision repair pathway, are critical [80]. In this regard, olaparib (Lynparza, AstraZeneca AB, Södertälje, Sweden) was the first drug approved by the FDA in 2018 for the treatment of patients with HER2-negative breast cancer and BRCA mutations [81], but it has many reported side effects [82]. Talazoparib (Talzenna, Pfizer Europe MA EEIG, Belgium) is another drug approved by the FDA in 2018, but for HER2-negative and locally advanced or BRCA-mutated patients. In vitro, it showed 200-fold greater antitumor results than other PARP inhibitors [83]. However, the list of associated side effects is equally extensive [84].

Table 2 shows the main targeted therapies for treating breast cancer employed today.

## 2. Role of the Ca^2+^-Signaling Pathway in Breast Cancer

Ninety-nine percent of the total body calcium is found in the body in mineral form as calcium hydroxyapatite (Ca_10_[PO_4_]_6_[OH]_2_) associated with hard tissues, such as bones and teeth, which also act as a reservoir and source of free calcium ions (Ca^2+^) that are essential for bodily and cellular physiological functions [85].

As a second messenger, intracellular Ca^2+^ levels increase as a stimulus–response reaction, with an allosteric regulatory effect on enzymes and proteins involved in signal transduction pathways and different cellular processes, such as gene activation, secretion, migration, division, differentiation, proliferation and cell death [86], as well as invasion, metastasis and acquisition of drug resistance [87]. In the 1940s, a decrease in calcium levels in epidermal carcinoma cells was observed for several weeks, followed by the transformation of these cells into malignant ones, when this precancerous condition was experimentally induced [88]. Since then, the central role of this ion and proteins involved in Ca^2+^-signaling pathways in carcinogenesis and tumor progression has been widely reported [89] in different types of malignancies, including breast cancer [61,90], which is why blocking calcium signaling was proposed as a promising strategy to improve the efficacy of current anticancer therapies, as well as antitumor immune responses.

Calcium homeostasis is achieved by keeping cytosolic calcium levels low, with the extracellular space, cytoplasm, endoplasmic reticulum and mitochondria being the four primary compartments involved in cellular Ca^2+^ circulation [91], and with both the mitochondria and endoplasmic reticulum serving as intracellular calcium stores. Indeed, in the face of an extracellular Ca^2+^ concentration of 1.3 mM [92], cytoplasmic Ca^2+^ in resting cells is maintained at concentrations ranging from 0.05 to 0.15 mM [93,94], mainly due to the coordinated function of calcium receptors, organelle and membrane ion channels, membrane pumps and transporters, as well as calcium buffer proteins. The modulation of the Ca^2+^ concentration is tightly regulated according to cellular needs [92] by three main processes that are not mutually exclusive [85]:Amplitude modulation [95]: the process responsible for triggering different downstream signaling responses, as proteins with higher Ca^2+^ binding affinity are activated at lower Ca^2+^ concentrations, whereas proteins with lower Ca^2+^ binding affinity are activated at higher concentrations [96].Frequency modulation: the process by which repetitive and transient increases in cytosolic Ca^2+^ concentration led to different protein activation [95].Modulation related to the spatial distribution of signals, which depends on the localization of effectors to Ca^2+^ modulators, such as channels [97].

Considering that the ion concentration in luminally mammary glands and breast milk is 10 mM and 2–4 mM, respectively, Ca^2+^ homeostasis is especially important in mammary gland cells, even more so during the lactation process, with them being very sensitive to changes in Ca^2+^ signaling, concentration and modulation mechanisms, which are also decisive in breast cancer progression [98]. However, despite the importance of Ca^2+^ during lactation and the association of dysregulation of calcium homeostasis and signaling with mammary gland pathophysiology, the implications of calcium signaling in the regulation of cell proliferation, differentiation and apoptosis are not yet fully understood [99].

### 2.1. Proteins Involved in Calcium Homeostasis and Relevance in Breast Cancer

Multiple proteins are directly involved in the regulation of cellular Ca^2+^ homeostasis, and thus, the cellular response. Alterations in Ca^2+^ channels, G-protein-coupled receptors (GPCRs), calcium buffer proteins and ATPases were described as hallmarks of different types of cancer and as potential drug targets for breast cancer treatment [99].

#### 2.1.1. Calcium Ion Channels

Calcium ion channels are transmembrane proteins with selective Ca^2+^ permeability that allow calcium to flow across cell membranes through a central pore. These ion channels are very diverse in both structure and function, with voltage-dependent calcium channels being one of the main types at the plasma membrane.

Voltage-dependent ion channels are integral membrane proteins that rapidly open and transport calcium to the cytoplasm upon electrochemical-gradient-driven changes in cell membrane voltage. They are widely expressed in neurons and muscles and play key roles in synaptic transmission and muscle contraction, respectively. Aberrant functioning of these channels was detected in different malignancies, such as melanomas and gliomas, as well as in prostate, colon, pancreatic and breast cancers [100]. Recent studies in this field reported that calcium channel subunit 4 (CACNG4), which is overexpressed in breast cancers with poor prognosis [100], is involved in cell proliferation, adhesion and invasion [101], and the potential utility of channel antagonists to inhibit cell proliferation and adhesion in breast cancer was suggested [100].

#### 2.1.2. Ligand-Dependent Calcium Ion Channels

Among ligand-dependent channels (LGCCs), Ca^2+^-release-activated channels (CRACs) are responsible for the following:Store-operated Ca^2+^ entry, which is a process that serves to replenish Ca^2+^ after its release from reserve sites, such as the endoplasmic reticulum.Cytosolic calcium increases are necessary for cell activation. The entire process of calcium-dependent cell activation is based on ER calcium release combined with capacitative calcium influx and consequent increases in cytosolic calcium levels.

The pores of LGCC channels are formed by plasma membrane ORAI proteins that work in concert with the endoplasmic reticulum stromal interaction molecule (STIM), which senses an ER luminal calcium decrease during Ca^2+^ mobilization and activates ORAI upon depletion of Ca^2+^ storage (Figure 2) [85]. Other types of LGCC channels, such as transient receptor potential channels (TRPC) that allow Ca^2+^ fluctuations (Figure 2), also play a major role in this process [102] by promoting membrane hyperpolarization and Ca^2+^ entry into cells [103]. Both types of LGCCs can bind to form heteromeric complexes for a major Ca^2+^ entry into the cell, which has been associated with poor prognosis in cancer patients [104,105].

#### 2.1.3. Voltage-Dependent Calcium Channels

Voltage-dependent calcium channels (VGCCs) comprise five subtypes: L, R, P/Q, T and N, with T-type channels playing a key role in regulating cytosolic calcium levels. These channels present three isoforms of the ∝_1_ subunit (Ca_V_1, Ca_V_2 and Ca_V_3) [106] that have each generated a subfamily. The Ca_V_3 isoform consists of three subtypes: Ca_V_3.1 (CACNA1G), Ca_V_3.2 (CACNAH1) and Ca_V_3.3 (CACNA1I), where their functions include the regulation of the G1/S checkpoint of the cell cycle [107] and programmed cell death [108], evidencing their importance in carcinogenesis. Although there is still no drug for T-type channels, the blockade of these channels appears to contribute to the therapeutic utility of other drugs, making them a new target for anticancer drug development [109].

#### 2.1.4. G-Protein-Coupled Receptors

G-protein-coupled receptors (GPCRs) also have an indirect role in the initiation of Ca^2+^ signaling upon activation by different extracellular signals. Ligand binding to these membrane receptors causes a change in receptor conformation that promotes the activation of cytoplasmic G proteins (Gα, Gβ and Gγ), which, in turn, can activate the membrane-associated enzyme adenylyl cyclase responsible for the second messenger cAMP from ATP molecules, as well as activation of phospholipase C that converts phosphatidylinositol-4,5-bisphosphate (PIP2) into the secondary messengers diacylglycerol (DAG) and inositol-1,4,5-trisphosphate (IP3). While DAG remains within the membrane, IP3 diffuses into the cell and interacts with its calcium receptor channel in the endoplasmic reticulum, promoting Ca^2+^ outflow from the lumen into the cytoplasm (Figure 2) [85]. It should be noted that various growth factor receptors, such as Epidermal Growth Factor (EGF), human Epidermal Growth Factor Receptor (EGFR), Transforming Growth Factor-alpha (TFG-α) and Platelet-derived growth factor (PDGF), can also induce calcium signaling via PLC-γ activation [110,111,112].

#### 2.1.5. Calcium Buffer Proteins

When Ca^2+^ enters the cell, it rapidly binds to negatively charged proteins, such as calbindin-D28k, calbindin-D9k, calreticulin, parvalbumins, calnexin, calretinin, GRP78/94 and calsequestrin, which act as effectors or buffers [113], transporting ions across cells and causing changes at the level of amplitude, frequency and spatial distribution that limit the availability of free Ca^2+^ to activate cellular functions, such as differentiation, transcription, migration, motility and phagocytosis [114,115,116]. Effectors, such as the annexin family of proteins, troponin C, calpain protease, myosin light chain kinase, synaptotagmin, nitric oxide synthases, cadmodulin-dependent protein kinase (CAMK), downstream regulatory element antagonist modulator (DREAM) and cyclic AMP response element binding protein (CREB), initiate downstream signaling pathways that ultimately induce activation of cellular functions [85].

Mitochondria rapidly internalize Ca^2+^ through the outer mitochondrial membrane, but to cross the inner mitochondrial membrane, they need the mitochondrial calcium uniporter complex (MCU) to accumulate Ca^2+^ in the mitochondrial matrix. In turn, to export it from the matrix, mitochondria release Ca^2+^ via a mitochondrial Na^+^–Ca^2+^ exchanger (NCLX) [117]. Although the most important intracellular Ca^2+^ stores are in the endoplasmic reticulum (ER), the mitochondrial Ca^2+^ concentration also influences cytosolic concentration by regulating cellular processes such as cell death by necrosis and cell apoptosis [118].

The concentration differences and the transport mechanisms involved in gradient maintenance are critical in Ca^2+^ signaling, which is a process that is essential for cellular homeostasis.

Alterations in the expression of Ca^2+^ channels, receptors and buffers can cause calcium levels to increase above the physiological threshold, promoting uncontrolled cell proliferation and the acquisition of a malignant phenotype [119] caused by transcriptional activation of genes that promote tumor growth. Although it might be thought that the decrease in Ca^2+^ levels or even its depletion could be a solution to stop the signaling pathways leading to the acquisition of this phenotype, this is not the case. The decrease induces tumor chemoresistance and evasion of cell apoptosis, which justifies the need for further studies in this field [120,121].

### 2.2. Ca^2+^ as a Therapeutic Target in Breast Cancer

Despite the worldwide effort to raise awareness of breast cancer and the improvement of detection and screening methods and treatment strategies, up to 5% of patients have metastases at the time of diagnosis, for which a complete cure is not possible, and up to 30% of women diagnosed with early-stage disease progress to metastatic breast cancer [122] due to intrinsic or acquired drug resistance; therefore, lines of research aimed at decreasing the high mortality rates in patients with metastatic breast cancer are priority areas.

A total of 75% of advanced breast cancer cases present with bone metastases, and 70% of them show pathological cancer-associated bone pain, bone resorption and microfractures, which significantly affect their quality of life [123]. Since both pathological breast-cancer-associated bone pain and breast calcifications share Ca^2+^ as a common component, there are currently different lines of research focused on identifying potential biomarkers of Ca^2+^-signaling pathways that can be used to treat breast cancer and to prevent bone pain, breast calcifications and tumor progression [124,125].

During routine mammography, it is common to detect breast calcifications, which are calcium deposits formed by different calcium salts, such as calcium oxalate and hydroxyapatite, with the participation of metals—such as zinc, magnesium and iron, where the latter is especially found in malignant calcifications [126]—within the breast tissue. Although breast calcification may be associated with different pathological processes, such as inflammation, infection or benign lesions, especially after the age of 50 years [127], they are present in about 30% of all malignant breast lesions, in more than 50% of malignant infraclinical breast lesions and in up to 85–95% of ductal carcinomas in situ [128] such that both the detection of microcalcifications in mammograms and their composition were proposed as risk factors for the development of breast cancer [129].

Although the pathophysiology of mammary calcifications is not well understood yet, they were reported to be caused by a combination of abnormal expression of bone matrix proteins and alterations in the secretory pathway of calcium ATPase (SPCA2) isoform [127], which is a Golgi-localized protein responsible for Ca^2+^ and Mn^2+^ sequestration required for proper protein folding, glycosylation and sorting from the RE to Golgi vesicles [130]. In humans, the SPCA1 and SPCA2 isoforms are encoded by the ATP2C1 and ATP2C2 genes, respectively, and differ from each other by their N-terminus, as well as by the higher affinity of SPCA1 for Ca^2+^ relative to SPCA2 [131]. Although the functions of both isoforms are being explored, studies show that while SPCA1 is elevated during the mid-lactation phase, SPCA2 is responsible for Ca^2+^ accumulation in the Golgi apparatus during lactation, especially just before parturition [131]. SPCA2 is frequently overexpressed in the tumors of patients with hormone-receptor-positive (ER+/PR+), which is associated with poor prognosis, as it exerts a pro-survival effect on mammary epithelial tumor cells. SPCA2 activates Ca^2+^ entry through ORAI1 channels via a constitutive mechanism called store-independent calcium entry (SICE), where it acts as a strong activator of the ORAI1 channel with its interaction with the N- and C-terminal domains, causing intense Ca^2+^ entry into the plasma membrane. This promotes cell survival, progression and chemoresistance of breast cancer cells [132]. However, silencing of SPCA2 expression increases mitochondrial ROS production, DNA damage and activation of the ataxia-telangiectasia-mutated/rad3 kinase–p53–related kinase (ATM/ATR) axis, which arrests the cell cycle in the G0/G1 phase and induces apoptosis. Hence, SPCA2 was proposed as a prognostic marker and its knockdown was proposed as a possible therapeutic potential in the treatment of breast cancer [132,133].

In line with these results, alterations in the expression of other Ca^2+^ channels have been associated with different breast cancer subtypes (Table 3).

Given the role of calcium channels in the regulation of the epithelial–mesenchymal transition (EMT), recent studies proposed the use of calcium channel blockers as a therapeutic strategy to inhibit EMT in cancer cells [132]. However, conflicting results showing both the association of aberrant expression of Ca^2+^ channels and pumps to triple-negative and hormone-receptor-positive breast tumors with poor prognosis [132,149], but also the better survival of patients with luminal subtype tumors warrant further studies in this area of research.

Studies in this field showed that Ca^2+^ pumps are highly elevated in breast cancer cells in a subtype-specific manner and that changes in their expression are often correlated with tumor progression [150] (Table 4).

Ca^2+^ signaling promotes reactive oxygen species (ROS) in mitochondria and the phosphorylation and translocation to the cell nucleus of signal transducer and activator of transcription 3 (STAT3), which is a transcriptional activator in breast cancer that regulates the activation of several target oncogenes associated with immunosuppression, malignant transformation, tumor growth, apoptosis, metastasis and chemoresistance [156]. Consistent with studies demonstrating that STAT3 is an early diagnostic tumor marker that is often constitutively overexpressed and activated in breast cancer, strategies aimed at modulating Ca^2+^ signaling in these tumors may be useful as a novel therapeutic approach.

Calcium signaling is also linked to mitogen-activated protein kinases (MAPK, MAPK/ERK, Ras-Raf-MEK-ERK), which are kinases involved in extracellular signaling transduction related to growth, proliferation, differentiation, development, transformation, migration, resistance and cell death, which are frequently overactivated in breast carcinomas [157]. For example, overexpression of SPCA2 in hormone-receptor-positive breast tumors results in the upregulation of SICE, which activates the tumorigenic MAPK pathway [133]. Similarly, TRPC3 acts as an anti-apoptotic regulator through the MAPK pathway [158]. MAPKs are tightly regulated by phosphatases and bidirectional communication with other kinases that regulate cell survival and proliferation, such as protein kinase B PKB/Akt, which is a serine/threonine protein kinase that is often dysregulated in breast cancer when abnormal Ca^2+^ signaling occurs. Given the important role of MAPKs and Akt in malignant breast cancer behavior and resistance to conventional treatments [157], targeting Ca^2+^ signaling using a channel blockade could also represent a useful therapeutic approach in tumors with such kinase alterations.

## 3. Preclinical and Clinical Research on Ca^2+^ in Breast Cancer

Although most studies have focused on evaluating the role of Ca^2+^-signaling pathways in tumor proliferation and/or identifying those channels with deregulated expression in breast cancer cells, different groups have gone a step further in this field.

Before starting clinical studies, drug development programs go through a preclinical phase in which both in vitro and in vivo research is carried out to investigate the possible therapeutic potential of a given candidate molecule to treat the disease. In vitro, one of the strategies followed to trigger tumor cell apoptosis has been based on the use of heavy-metal-based drugs to increase intracellular calcium levels [159,160]. Specifically, one study measured the effects of the gold compound auranofin on cell apoptosis and intracellular Ca^2+^ concentration in MCF-7, showing that this drug increases the Ca^2+^ concentration, although the origin of the increase could not be determined when trying to block different receptors [161]. Given the risk of drug resistance and associated toxicities, the potential use of this type of drug is very limited [162,163].

On the other hand, melatonin, which is a hormone that regulates the calmodulin-mediated Ca^2+^-signaling pathway through G-protein-coupled membrane receptors, was also shown to change the level of intracellular Ca^2+^ concentration. One study in this field determined that while ATP can induce MCF-7 cell growth, melatonin can abrogate MCF-7 cell proliferation and that pretreatment with melatonin followed by ATP in MCF-7 cells can further suppress cell proliferation [164], which deserves additional research.

Baicalein, which is a natural polyphenolic pigment, was also shown to induce apoptosis in breast, gastric, prostate and hepatoblastoma cancer cells [165,166,167,168], which has motivated in vitro studies on the role of Ca^2+^ and its signaling pathway in apoptosis induced by this pigment. The results for MDA-MB-231 show that baicalein has effects on apoptosis through the inhibition of antiapoptotic Bcl-2, induction of proapoptotic Bax proteins and caspase-3 [169].

Recently, electroporation has been incorporated as a novel therapeutic approach for cancer treatment with less risk of adverse effects than conventional treatments, such as surgery or radiation, and greater durability of effect and requiring less cost [170,171]. Preclinical results showed that lipid composition and heat capacity influence cell permeability [170], which can be used to facilitate the transportation into tumor cells of chemotherapeutic drugs, such as bleomycin or cisplatin, to increase their cytotoxicity. Electrochemotherapy is understood as the permeabilization of tumor cells by electroporation after intravenous injection of a chemotherapeutic drug, commonly bleomycin (Figure 3). The first clinical trial with electrochemotherapy was performed in 1990 [172], and since then, many trials have been performed to treat breast tumors [173,174]. More recently, in 2018, the first clinical trial with Ca^2+^ electroporation demonstrated the utility of this electroporation modification in which supraphysiological doses of Ca^2+^ were used after electrochemotherapy as an effective and safe anticancer treatment [175]. Ca^2+^ electroporation treatment is a modification of conventional electrochemotherapy because, after electroporation, supraphysiological doses of Ca^2+^ are used (Figure 3) [176]. Given its favorable cost–benefit ratio, Ca^2+^ electrochemotherapy has turned into a promising therapeutic approach that is no less effective than conventional electrochemotherapy [171].

Given the lack of major hormone receptors and the limited number of therapeutic options for TNBC patients, electroporation represents a promising option [177], having already demonstrated its palliative effect by reducing patients’ pain [178,179].

Clinical cases in which patients with HER2-positive breast cancer skin metastases were treated using (1) trastazumab alone, (2) trastazumab emtansine (TDM1), or (3) a combination of trastazumab and Ca^2+^ electroporation showed that although TDM1 was more effective on skin metastasis than trastazumab alone, the side effects associated with TDM1 were not well tolerated. On the other hand, the study also showed that a combination therapy of transtazumab and Ca^2+^ electroporation intermittently applied when needed effectively controls metastasis during the applied period with better tolerance to the chemotherapeutic, including a better preservation of the skin area in which the Ca^2+^-electroporation was applied, which justifies continuing the investigation in a phase II study [180].

Failure of anthracycline chemotherapy, one of the most widely used cytotoxic drugs for breast cancer treatment, is often associated with the poor prognosis of patients, as “salvage” chemotherapy usually has a low response rate. The use of verapamil, which is a potent Ca^2+^ channel blocker commonly used to treat hypertension, was shown to increase survival in patients with metastatic breast cancer with anthracycline resistance [181]. Moreover, verapamil also has important effects on drug efflux pumps of the ABC transporters family involved in cytotoxic drug resistance. Although these results were promising, the use of calcium channel blockers has always raised great doubts as to whether they could contribute to tumor growth by inhibiting Ca^2+^-signaling-mediated apoptosis and thereby inducing cell growth. In this regard, some recent studies found no evidence that long-term exposure to Ca^2+^ channel blockers is associated with an increased risk of breast cancer [182], but not all agree on this, which still generates much uncertainty, especially in the long term and depending on the subtype of breast cancer [183].

## 4. Conclusions

Although the treatment of breast cancer has been improving over the last few decades, there are still numerous cases of patients who die because of this disease. This has generated the need to identify new therapeutic targets that allow, on the one hand, for improving patient survival and, on the other hand, understanding the mechanisms underlying the current resistance to existing therapeutic agents.

The fact that both prolonged elevation and depletion of intracellular Ca^2+^ is oncogenic in nature, as well as deregulated expression varies according to breast cancer subtype, has led to the need to better understand the specific molecular mechanisms driving the acquisition of this malignant phenotype [119,120,121].

Specific and selective targeting of the Ca^2+^-signaling pathway could be an important approach in the precision medicine of breast cancer treatment, something already evidenced in some in vivo studies. However, much remains to be done in this field and many more studies are required to optimize the therapeutic strategies to be followed in the clinical practice of this disease, including the consideration of the possible pharmacological interactions that can be produced by changes in the calcium-signaling pathways.

## Figures and Tables

**Figure 1 cancers-15-02872-f001:**
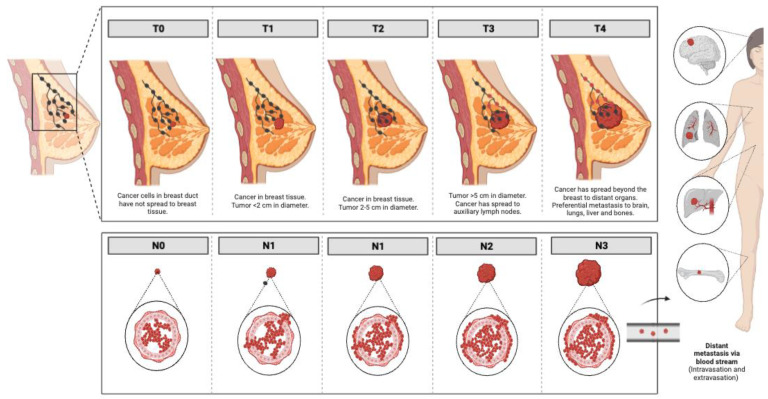
Progression of breast cancer in the different stages according to the traditional TNM staging system. Images were created using Biorender.com.

**Figure 2 cancers-15-02872-f002:**
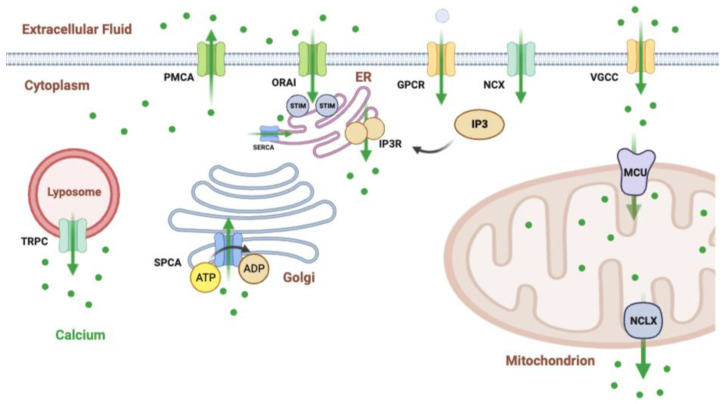
Ca^2+^ channels and pumps involved in cell homeostasis. Images were created using Biorender.com.

**Figure 3 cancers-15-02872-f003:**
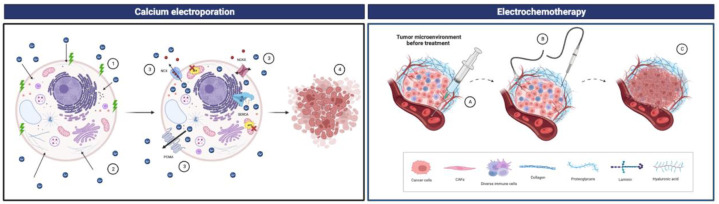
Conventional electrochemotherapy ((A) intravenous injection, (B) application of electrical pulses and (C) tumor cell death) versus Ca^2+^ electroporation ((1) application of electrical pulses, (2) supraphysiological calcium doses, (3) calcium transport and (4) tumor cell death). Images were created using Biorender.com.

**Table 1 cancers-15-02872-t001:** Types of immunotherapeutics used to treat breast cancer.

Type	Molecular Target	Immunotherapeutics	Type of Patient
Tyrosine kinase inhibitors (TKIs)	ATP-binding side of HER2 and EGFR tyrosine kinase	Lapatinib	HER2+
	Tyrosine kinase domain of HER2	Neratinib	HER2+ treated with adjuvant Trastazumab therapy
	Tyrosine kinase domain of HER2 and HER3	Tucatinib	Advanced-stage HER2+
	Human epidermal growth factor receptor type 1 (EGFR) specific tyrosine kinase domain	Pyrotinib	Advanced-stage HER2+ previously treated with chemotherapy
HER2	Extracellular domain of HER2	Trastazumab	HER2+
		Pertuzumab	HER2+ with risk of relapse
		Margetuximab	HER2+ in metastasis stage
Drug–antibody conjugates	Extracellular domain of HER2+ microtubule depolymerizer	Trastazumab emtansine (T-DM1)	HER2+ in metastasic stage
	HER2 extracellular domain + maleimide + topoisomerase inhibitor	Trastazumab deruxtecan (DS-821a)	HER2+

**Table 2 cancers-15-02872-t002:** Types of targeted therapies against breast cancer.

Type	Molecular Target	Chemotherapeutic	Type of Patient
Angiogenic	VEGF-A	Bevacizumab	HER2-
CDK4/6 inhibitor	CDK4/6	Palbociclib Ribociclib Abemaciclib	HER2+ or HER2-
PI3K/Akt/mTOR pathway inhibitors	PI3K	Alpelisib	Hormone-receptor-positive and HER2-positive
	mTORC1	Everolimus	Advanced hormone-receptor-positive and HER2-positive
PARP inhibitors	PARP	Olaparib	HER2- and BRCA-mutated breast cancer
		Talazoparib	HER2- and locally advanced tumors or with mutations in BRCA

**Table 3 cancers-15-02872-t003:** Alteration of the calcium signaling pathway in breast cancer.

Ion Channel	Member	Overview
ORAI protein	ORAI1 and ORAI3	Higher expression levels of the ORAI1 isoform in hormone-receptor-negative subtypes and ORAI3 in hormone-receptor-positive subtypes [134]. ORAI3 was found to be overexpressed in 76.9% of breast cancer samples analyzed [135]. ORAI1 increases in expression during lactation [136]. ORAI1 and ORAI3 have been established as therapeutic targets in hormone-receptor-positive and hormone-receptor-negative breast cancers, respectively [137]. ORAI1 regulates the stimulation of the SICE (store-independent calcium entry) pathway and ORAI3 initiates the SOCE (store-operated calcium entry) pathway.
STIM protein	STIM1 and STIM2	STIM1 shows higher expression in hormone-receptor-positive patients [102], which is associated with increased aggressiveness and worse prognosis [138]. STIM1, STIM2 and ORAI3 mediate the SOCE pathway in the MCF-7 cell line, while STIM1 and ORAI1 mediate the SOCE pathway in the MDA-MB-231 cell line [139]. Although the origin of these differences is unknown, it appears that sex hormones play a key role in regulating the expression of the different ORAI isoforms in breast cancer [140]. High STIM1 and low STIM2 phenotypes are associated with the basal subtype and correlate with a worse prognosis [137]. The SOCE pathway is mediated by STIM1/2 and ORAI3 in ER-positive breast cancer cells whereas ER- cells use the STIM1 and ORAI1 pathways [140].
Transient Receptor Potential Canonical (TRPC) channels	TRPC6	Abnormal expression of TRPC6, along with TRP melastin (TRPM) and TRP vanilloid (TRPV) channels, is observed in breast cancer [141,142,143]. TRPC6 allows for the translocation of ORAI isoforms to the plasma membrane [144].
Channel TRP	TRPC1 and TRPM2	In the MDA cell line, silencing of TRP channels blocks the ability to express the EMT marker vimentin, which allows us to intuit different Ca^2+^ influx pathways responsible for the epithelial–mesenchymal transition (EMT). The TRPC1 channel has a higher expression level in the TNBC subtype than in luminal A, luminal B or HER2+ subtypes [145]. Overexpression of the TRPM2 channel in luminal B patients and low expression in HER2+ patients evidenced worse patient outcomes [146].
VGCC type T	CA_V_3.2 and CACNA1G	Ca_V_3.2 isoform can be used as a marker in luminal A, luminal B and HER2-enriched subtypes versus basal subtypes. High levels of Ca_v_3.2 were associated with worse outcomes in ER+ patients. However, high levels are positively associated with survival after chemotherapy in HER2+ patients [147]. 7 VGCC family members (CACNA1C, CACNA1D, CACNA1A, CACNA1B, CACNA1E, CACNA1H and CACNA1I) were shown to be underexpressed in breast cancer [148]. Ca_V_3.1 (CACNA1G) can be used to distinguish invasive versus mucinous lobular breast cancer [148].

**Table 4 cancers-15-02872-t004:** Altered Ca^2+^ pumps in breast cancer.

Ca^2+^ Pumps	Member	Overview
SERCA-ATPases of the endoplasmic reticulum	SERCA3	Their expression is greatly decreased in precancerous lesions and inversely correlated with tumor grade in triple-negative invasive breast tumors compared with receptor-positive tumors [150]. SERCA3 increases in response to TGF-β (tumor growth factor beta) during the epithelial–mesenchymal transition of tumor cells. It was proposed that a loss of SERCA3 may be implicated in a loss of an IP3-mobilized endoplasmic reticulum compartment, thus altering the ability to respond to stimuli through IP3 [150].
ATPases of the plasma membrane	PMCA2	Recent studies reported the role of plasma membrane calcium pump isoform 2 (PMCA2), which is expressed in human epithelia undergoing lactational remodeling, in Ca^2+^ efflux from mammary cells into milk [151] but also during carcinogenesis and tumor progression, where it was found to be overexpressed in up to 9% of human breast cancers [130]. Overexpression of PMCA2 has been associated with poor prognosis in triple-negative breast cancer patients younger than 50 years, as well as with poor survival in patients with HER2-positive tumors [130], apparently due to the interaction between the HER2 receptor and PMCA2 at actin-rich sites in the plasma membrane, where the Ca^2+^ pump maintains the ion concentration at low levels affecting Ca^2+^ homeostasis [152].
	PMCA4	The role of plasma membrane calcium pump isoform 4 (PMCA4) inhibition has been associated with Bcl-2 inhibitor ABT-263-mediated apoptosis, as well as NF-kB-induced promotion of cell death in MDA-MB-231 breast cancer cells. The fact that breast cancers with a poor prognosis are associated with elevated constitutive NFkB activity makes them a potentially effective tool in the therapy of this disease [153]. In BRAF-mutated melanomas, PMCA4b was associated with increased expression of the pump, inhibiting the migratory, and thus, the metastatic capacity of the cells [154]. Studies in which the differentiation of MCF-7 breast cells was induced by treatment with histone deacetylase inhibitors (HDACis) showed an increase in PMCA4b expression. Increased PMCA4b expression leads to Ca^2+^ clearance in cells, contributing to normal mammary epithelium development, and thus, to tumor cell elimination [155].

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
