# Peer review of "Calcium Homeostasis in the Development of Resistant Breast Tumors"

_cancers, 2023, doi:10.3390/cancers15112872_

Round 1

Reviewer 1 Report

Reviewer’s comments to « Influence of alterations in calcium homeostasis on the development of resistant breast tumors” by Martin-García et al.

Breast cancer is a complex disease in which several molecular oncogenic mechanisms, various premalignant lesion types and lines of differentiation are involved. In addition, various breast cancer types may present specific vulnerabilities that can be targeted by specific types of treatments. Various breast cancer types are recognized, and these can display important differences in terms of prognosis and response to treatments. Moreover, the evolution of tumors under the pressure of various types of treatments can lead to the emergence of several resistance mechanisms. A full understanding of breast molecular oncogenesis and tumor evolution would be important for the development of curative therapies, in particular for inoperable/disseminated disease. On the other hand, a significant amount of information has been accumulated regarding the involvement of cellular calcium signaling in normal and tumoral breast epithelial biology, and various types of cross-talk between calcium signaling and breast carcinoma cell behavior, proliferation, differentiation, motility or apoptosis are currently being discovered. The study of the interactions between calcium signaling and breast carcinoma constitutes thus an important field of research.

In this work, Authors give a short overview of some basic notions of breast cancer biology such as tumor heterogeneity, progression, histological and molecular types, dissemination and therapy. Thereafter Authors briefly discuss various aspects of intracellular calcium signaling, and its role in normal mammary gland biology (lactation) and basic protein types of the calcium homeostatic toolkit (calcium binding proteins, channels and pumps). Authors then discuss our current knowledge about changes of the expression and the activity of various calcium channels and pumps observed in various cellular models of breast cancer, and mention different pharmacological approaches and experimental treatment modalities such as  therapeutic electroporation that are based on calcium-related cellular mechanisms.

Comments:

A major problem with this Manuscript is that basic notions used in the characterization and classification of breast carcinomas seem to be used apparently erroneously. This precludes publication of the Manuscript in its present form, because this would mislead the readers. The Reviewer believes that the notion of tumor stage and tumor grade are used imprecisely. For example, in the paragraph “Heterogeneity of breas (sic) cancer: origin, progression and molecular staging”, it is not clear, what “molecular staging” is supposed to mean. Tumor stage and tumor grade are very rigorously and clearly defined notions that cannot be used interchangeably, these are separate distinct notions. Authors should explain, what they mean by “molecular staging”. In the first part of this paragraph tumor staging (i.e.: for example, the progression in time from localized to disseminated disease) is discussed without reference to molecular features and this is followed by a discussion of various molecular subtypes of the disease (and their prognostic a predictive implications). The notions of molecular (and histological) tumor grade, stage/progression and molecular, as well as transcriptome-based classification need to be presented clearly and unambiguously, because these notions constitute currently the basis of the diagnostic workup of breast cancers, and are fundamental for breast cancer research as well.

Another problem in this paragraph is related to the discussion of tamoxifen. It is stated that “tamoxifen … by competitive inhibition decreases circulating estrogen levels”. It is not clear, how a competition, by tamoxifen, with estrogen would decrease circulating estrogen levels. Competitive inhibition probably decreases the interaction between estrogen and estrogen receptors, and would thus decrease downstream estrogen receptor signaling. However, how this relates to decreased circulating estrogen levels is not clear.

Authors also state that “Overall, the 5-year overall survival is 85% for TNBC versus 94-99% for hormone receptor-positive and HER2-positive breast tumors” : is the 94-99% five years overall survival valid regarding, for example, hormone-receptor-positive metastatic disease, as well? Please explain the meaning of the two instances of “overall” and clarify.

Authors also state in this paragraph that “ On the other hand, breast tumors lacking ER, PR and HER2 are called basal-like and are a group of tumors characterized by high expression of cytokeratin 5/6 (CK5/6) and/or HER2 marker (lanes 89-91).” Do Authors mean that tumors lacking HER2 are characterized by high expression of HER2 marker?

These issues give the strong impression to the Reviewer that this part of the Manuscript needs to be rewritten in depth, and precision improved significantly.

Please note that, in the opinion of the Reviewer, statements in lanes 55-60 are also rather debatable; in particular, the proposed association of ductal and lobular carcinomas with ducts and acini is problematic, as current consensus suggests that these cancers arise in the terminal ductal-lobular units (TDLU). The Reviewer believes that this part of the Manuscript would benefit from updating, based, for example, on publications such as: Breast Tumours. WHO Classification of Tumours, 5th Edition, Volume 2. WHO Classification of Tumours, ISBN-13 : 978-92-832-4500-1.

Serious errors can be seen also in Fig. 2 : Please note that PMCA enzymes transport calcium ions from the cytosol into the extracellular space, and that EGFR is not a calcium channel, and that the name of the Orai channel is missing. In addition, if SPCA in Golgi is depicted, it would be appropriate to show SERCA enzymes in the ER as well.

Lane 375 : Please note that the name of SPCA is “secretory pathway calcium ATPase”; this is a calcium pump, not a “calcium-secreting protein”.

In Table 4 it would be appropriate to discuss literature also on PMCA4 and breast cancer.

In conclusion, the Reviewer believes that a careful in-depth improvement process needs to be undertaken before this work can be considered for publication, because the Paper in its present form contains several errors and bits of imprecise and potentially misleading information for non-expert readers. Given the extent of these, Authors should consider the radical revision of their Manuscript or a new submission of their improved work altogether.

English is OK, but it is possible that some subtle errors of syntax lead to misunderstanding.

Reviewer 2 Report

The manuscript cancers-2367747, entitled “Influence of alterations in calcium homeostasis on the develop-2 ment of resistant breast tumors” by Martin-García and coworkers discusses the role of Ca2+ in physiological and pathophysiological conditions, addressing both “in vitro” and “in vivo” studies. The authors elegantly and thoroughly gather the current knowledge about Ca2+ homeostasis and its role in the pathophysiology of breast cancer. The ms is certainly compelling and timely, as several of the proteins involve in the machinery that fine-tune Ca2+ homeostasis are being targeted by novel therapies against breast cancer.

I believe that the ms is suitable to be accepted for its publication in Cancers.

Minor points:

Page 3, Line 83, “15-10%” should be modified as “10-15%”.

Page 4. Lines 165-167, the sentence “Its approval in 2018 by the 165 FDA was for HER2-positive patients in combination with other anti-HER2 agents such as 166 trastuzumab” is confusing. The authors should rewrite it.

Page 7. Lines 303 and 307, change “storage” by “store”.

Page 9. Line 381, “studied, studies” sounds redundant, change it accordingly.

Page 9. Line 391, modify “Ca2+” by “Ca2+

Reviewer 3 Report

This manuscript was designed to review the role of calcium homeostasis in the development of breast cancer. Several Ca2+ influx pathways play an important role in regulating Ca2+ signaling in normal and breast cancer cells. The review summarizes our current understanding about the topic. Below is a list of issues that I considered should be addressed before the manuscript is published in Cancers.

1)     I would recommend the authors modify the title of the manuscript. “Influence of alterations..” does sound cumbersome.

2)     The authors provided an excellent introductory description of the pathophysiology as well as current and targeted therapies for breast cancer. However, they handle the role of Ca2+ signaling in breast cancer only superficially; in a manner that does not add anything significant to our current understanding of the disease. For example, in the section about Ca2+ ion channels, there is no mention of the role of T-type Ca2+ channels in breast cancer (references about this topic, including Ohkubo & Yamazaki, Int J Oncol. 2012; Taylor et al., Cancer Lett. 2008, Pera et al., Cancer Cell Int. 2016; Bhargava & Saha, Breast Cancer Res Treat. 2019 were not found in the references). Similarly, changes in ORAI and STIM proteins and its relevance in regulating Ca2+ signaling in breast cancer is lacking.

3)     Based on the issues raised in #2, the question we need to ask is: does the current review offers something new and compelling compared to previous work on the subject of Ca2+ signaling in breast cancer (see previous reviews by O'Grady & Morgan, Semin Cancer Biol. 2021; So et al., Semin Cell Dev Biol. 2019; Azimi et al., Br J Pharmacol. 2014 and many others). I would recommend that the authors carefully analyze the objective of the review and focus the information provided.

4)     The authors should describe with more examples how disruption of Ca2+ signaling alters the progression of breast cancer (for example, how does “blocking calcium signaling [Pag. 6, line 252] alter cell proliferation/cell death, metastatic potential, or epithelia-mesenchymal transition).

5) I am always puzzled by the citation of previous reviews in a review manuscript. The authors cited numerous reviews in the list of references (see references 79, 80, 94, 110, 112, and many others). I firmly believe that in a review, authors should focus on primary findings and putting those findings in an overarching manner that provides new insight into the scientific question discussed.

6) There are numerous grammatical errors throughout the text (proofreading by a native English speaker is recommended):

a)     Page 1, line 41: before progression to a metastatic disease

b)     Page 1, line 44, missing comma (this is a recurrent issue throughout the text): in autosomal dominant genes, such as 44 breast cancer 1…

c)     Page 2, line 47, unclear sequence: the ataxia telangiectasia 46 gene (ATM) and partner and localize of BRCA2 gene (PALB2) or low risk genes such as 47 single nucleotide polymorphisms …

d)     Page 2, line 51-52, missing commas: Moreover, different sociodemographic characteristics, such as lack of education, presence of anxiety or depression, or above-average comorbidities have been found …

e)     Page 2, line 67, missing comma: to the rest of the body's organs such as the bones …

f)      Page 4, line 136, missing comma: tamoxifen is replaced by aromatase inhibitor drugs, such as anastrozole b …

g)     Page 4, lines 147-148: In 1998, the FDA approved Trastuzumab, the first monoclonal antibody to treat HER2-positive patients.

h)     Page 4, line 166, missing comma: with other anti-HER2 agents, such as trastuzumab …

i)      Page 6, line 246, missing comma: different cellular processes, such as gene activation …

j) Page 7, line 304, missing comma: after its release from reserve sites such as the endoplasmic reticulum

see comments for the authors.

Round 2

Reviewer 1 Report

Comments to the revised version

Authors addressed several issues raised by the Reviewer. However, important points remain, as follows:

Title : « Alterations in calcium homeostasis on the development of resistant breast tumors” : alterations … “on” ? In the opinion of the Reviewer, the title is not very intelligible in its present form.

Lane 54 : Please note that parity and age at menarche etc. are also involved in breast cancer risk (for example see PMID: 23084519, PMID: 30867002).

Please add year of publication etc. for Ref.11.

Lane 63 : Please note that “carcinoma in situ” is not a site-dependent notion, such lesions can arise anywhere in the mammary epithelium. The various specialized notions used in pathology need to be employed with precision. In general, the Manuscript should be read and corrected therefore, by, for example, a clinical pathologist expert in breast cancer, before publication. Otherwise the clinical-pathological part of the Paper could be deleted or considerably shortened. For example : lane 77-78 : “Ultimately, breast cancer becomes invasive as it spreads to the rest of the body's organs” : here “metastatic” would be more adequate than “invasive”, as cancer can be locally invasive as well (independently of metastatic status).

Lane 73 : “anatomopathological characteristics of the tumor in different stages or stages.” : ?

Lanes 93-94 : “…and growth factors (human epidermal growth factor ErB2 receptor aka human epidermal receptor 2 HER2)” : Please rephrase, as ErB2 is not a growth factor.

Lane 98 : What is “low mitotic activity and prognosis” ?

Lanes 93 versus 116: Please explain the distinction between HER2 and EGRF.

Lanes 139-140 : “In such manner, therapies based on the use of monoclonal antibodies have revolutionized the treatment for HER2-enriched breast tumors” : please rephrase, and separate antiestrogen-based therapy from HER2-directed immunotherapy more clearly here.

Lane 145 : “of this tumors” : of these tumors

Lanes 146-147 “The 5-year overall survival for TNBC stage-I patients is 85%, compared to 94-99% of patients with hormone receptor-positive and HER2-positive breast tumors” : is this true for disseminated disease? 99% of disseminated/metastatic breast cancer patients survive for >5 years (see lane 128)? Please state stage also for hormone receptor-positive and HER2-positive breast tumors.

Importantly, please note that “molecular classification” nowadays is based also on more advanced transcriptomic analyses (Oncotype DX, MammaPrint etc.); this should also be discussed briefly.

Lanes 174-175 : “that ensure the complete elimination malignant cells” : …elimination of malignant cells

Lane 205 : “whose use in combination with chemotherapy…” : the use of which in combination with chemotherapy… (idem lane 208)

Lane 210 : “Kinase-dependent inhibitors” : no, these are not “kinase-dependent” inhibitors, these are kinase inhibitors (inhibitors of kinases).

Lane 229 : “and triggers”

Lanes 283-286 : “recent phase III results have shown that adding this drug does not improve the efficacy of treatment with paclitaxel (Abraxane, Bristol-Myers Squibb Pharma, Ireland) leading to the belief that if it were combined with estrogen receptor drugs the benefits obtained could be improved.” : Please explain.

Lanes 309-310 : “Indeed, since the discovery of the relationship between cancer and Ca2+ in the 1940s…” : the Reviewer is curious, what were the pertinent observations made on calcium and cancer in the 1940s. Please briefly explain these original observations and add original References.

Lane 323 : “Ca2+concentration” : Ca2+ concentration

Lanes 341-345 : “Although multiple proteins are directly involved in the regulation of cellular Ca2+ homeostasis and thus cellular response, alterations in Ca2+ channels, G protein-coupled receptors (GPCRs), calcium buffer proteins, and ATPases have been described as hallmarks of different types of cancer and as potential drug targets for breast cancer treatment.” : Why “although”?

Lane 347 : “Calcium ion channels are cell membrane-bound proteins” : these proteins are better describes as “transmembrane proteins”.

Lanes 349-350 : “with voltage-dependent calcium channels being the main type at the plasma membrane.” : in the opinion of the Reviewer this is incorrect; as also discussed by the Authors later, other types of receptor-coupled, or Orai-type etc. calcium channels are also present and relevant, in particular in non-excitable cells.

Lanes 359-360 : “and the potential utility of channel antagonists to inhibit cell proliferation and adhesion in breast cancer.” : Please rephrase (“and the potential utility of channel antagonists to inhibit cell proliferation and adhesion in breast cancer has been suggested”) or similar (predicate is missing).

Lanes 362-364 : “Among ligand-dependent channels (LGCCs), Ca2+-release-activated channels (CRACs) are responsible for store-operated Ca2+ entry, a process that serves to replenish Ca2+ after its release from reserve sites, such as the endoplasmic reticulum.” Please note that CRACs are also the source of cytosolic calcium increases required for cell activation, and their role is not limited solely to ER calcium replenishment. The entire process of calcium-dependent cell activation is based on ER calcium release combined with capacitative calcium influx (and consequent increases of cytosolic calcium levels).

Lanes 366-367 : Please specify that STIM “senses” ER luminal calcium decrease during calcium mobilization.

Lane 369 : “outflow of other cations, such as sodium or potassium” : Considering the fact that there is a strong inward sodium ion concentration gradient at the plasma membrane, the idea of Na+ outflow is problematic.

Lanes 371-372 : “Both types of LGCCs can bind to form complexes for a major Ca2+ entry into the cell”: Please explain the nature of these complexes (homo-/heteromers etc?).

Lane 380 : The notion of induction of necrosis via calcium channels needs to be explained in more detail, because, contrary to, for example apoptosis, necrosis is usually considered a passive process.

Lanes 385-394 : It should be noted that various growth-factor receptors can also induce calcium signaling (via PLCgamma activation).

Lanes 403-404 : Please note that DREAM is not a growth factor, but an intracellular, calcium-regulated transcriptional repressor. Please include specific References for DREAM function.

Lane 407 : “These concentration differences and the transport mechanisms…” : why “these”?

Lane 410 : “Figure 2. Ca2+ channels involved in cell homeostasis.” This title is incorrect, because pumps are also depicted. Also: TRP or TRPC? What is NCLN? Please enhance the discussion on mitochondrial calcium transporters.

Lanes 411-417 : Please rephrase for enhanced clarity.

Lane 445 : As already pointed out earlier by the Reviewer, the correct name is “secretory pathway calcium ATPase” (see for example :  PMID: 12543090, see also Ref. 126).

Lane 454 : “hormone-positive” : ?

Lane 457 : Please explain the mechanism whereby SPCA2 is coupled to calcium entry through ORAI1 in the plasma membrane.

Lanes 467-471 : “However, conflicting results showing both the association of aberrant expression of Ca2+ channels and pumps to triple-negative and hormone-positive receptor breast tumors with poor prognosis [126, 144], but also, to the better survival of patients with luminal subtype tumors warrant further studies in this area of research.” Please clarify and add References here for the association of aberrant expression of channels and pumps with better survival with luminal subtypes of tumors.

Table 4 : The current Ref.146 is not appropriate when Authors state that “Their (i.e.: calcium pump) expression is greatly decreased in precancerous lesions and inversely correlated with tumor grade in triple-negative invasive breast tumors compared to receptor-positive tumors “. The current Ref. 146 is a TRPC-related article.

Lanes 485-487 : “Calcium signaling is also linked to mitogen-activated protein kinases (MAPK, MAPK/ERK, Ras-Raf-MEK-ERK) : Please discuss the exact molecular mechanism/point of cross-talk/key protein interactions whereby calcium signaling regulates the MAPK pathway, and include additional References, the current Ref.153 (on ORAI and Akt) is not really adequate or sufficient in lane 489. Please specify (and include relevant References), exactly which calcium-regulated signaling protein activates which member of the MAPK pathway?

Lane 521 : “has also shown to induce” : “has also been shown to induce”?

Lanes 521-525 : It is not clear, how baicalein is coupled to calcium fluxes.

Lane 539 : Please delete second “an”.

Lanes 567-578 : Please note in text that verapamil has important effects also on drug efflux pumps of the ABC transporter family involved in cytotoxic drug resistance.

Lanes 585-586 : “The fact that both, prolonged elevation, and depletion of intracellular Ca2+ is oncogenic in nature” : please add relevant References here.

Lane 589 : “targeted targeting” : please rephrase.

Minor language editing would enhance clarity.

Author Response

Consulte el archivo adjunto.

Reviewer 3 Report

The quality of the manuscript has been been greatly improved. Minor revisions for grammar and spelling are recommended.

There are still some persistent grammatical errors:

1) Page 4, line 136, line 150, etc. missing comma:.. the production of growth factors, such as Epidermal..

2) Page 14, line 539, remove extra an...after electrochemotherapy, as an effective...

Please see suggestions for the authors.

Round 3

Reviewer 1 Report

Authors addressed most of the issues raised by the Reviewer, however, some improvement is still required as follows:

Lane 21 : “breast cancer being the most diagnosed malignancy” : …the most frequently diagnosed...? 

Lane 35 : “that by 2040 there will be 28.9 million new cases” : annually or cumulatively?

Lane 36 : “Although breast cancer mainly affects women, it has become the most diagnosed malignancy in the general population” : please rephrase; of course breast cancer mainly affects women; it is the most diagnosed malignancy in the general population also because of the number of cases in women. The phrase in its present form erroneously suggests that male breast cancer plays a significant role in determining breast cancer frequency in the “general population”. And it would be nice to briefly mention that although breast cancer is frequent and potentially fatal, early diagnosis and treatment lead to long-term remission/cure.

Lane 47 : “mutations in low-risk genes, such as single nucleotide polymorphisms” : Can SNPs be considered as mutations?

Lane 58 : “based on the site from which the tumor originated within the mammary gland as carcinoma in situ” : carcinoma in situ is not a site-dependent notion, such lesions can arise anywhere in the breast epithelium. Please rephrase.

Lane 88 : “This have highlighted” : This has highlighted (or : “These have highlighted)

Lanes 68-77 : It is not necessary to present the TNM classification of breast carcinoma in detail in this Paper, please shorten and summarize this part.

Lane 94 : “and human epidermal receptor 2 (HER2):” : human epidermal growth factor receptor 2

Lane 113 : “the absence of low levels of ER/PR expression” : absence of low levels meaning presence of high levels, or absence of even low levels?

Lanes 112-118 : “Basal-like tumors are characterized by the lack of HER2 overexpression and the absence of low levels of ER/PR expression. Triple negative breast tumors, lacking both hormone and HER2 receptors, constitute approximately 80% of basal-like tumors and 10-15% of breast carcinomas. These tumors are characterized by high expression of cytokeratin 5/6 (CK5/6) and/or expression of the human Epidermal Growth Factor Receptor (EGFR) family member HER2 (Human Epidermal Growth Factor Receptor 2/ErbB2 receptor tyrosine kinase 2) overexpression [23].” So Authors intend to state that tumors lacking HER2 are characterized by expression of HER2?

Lane 123 : “Insulin-Like” : Insulin-like

Lane 124 : “Tumor Growth Factor-alpha (TGFa)” : Transforming growth factor alpha?

Lane 144 : “when this is not clear based on traditional clinicopathologic features” : …clearly based on…

Lane 146 : “In.” or “Inc.”?

Lane 181 : “a selective modulator” : a selective modulator of what?

Lane 183 : “tamoxifen is replaced by aromatase inhibitor drugs” : it would be nice, if the pharmacological target of tamoxifen and aromatase inhibitors (i.e.: estrogen signaling) could be stated here.

Lane 196 : please delete “later”.

Lane 199 : “Recently, in 2021” : please delete “recently” or “in 2021”.

Table 1 : “Tyrosine Kinase-dependent inhibitors” : as already pointed out earlier by the Reviewer, these are not tyrosine kinase-dependent inhibitors, but “tyrosine kinase inhibitors”; inhibition does not depend on a tyrosine kinase, rather, the kinase is inhibited by the inhibitor.

Lane 233 : “as it depends on…” : as tumor formation depends on…

Lane 243 : Please explain “mono-phosphorylating” or delete.

Lane 296 : Please replace “[doi: 10.1126/science.99.2569.245-a]” by a Reference number.

Also, some more explanation is necessary here regarding decreased calcium levels in mouse epidermal carcinogenesis : was this observed in the skin within tumor cells or in the extracellular space surrounding tumor cells or in the adjacent normal tissue or in whole skin? How was carcinogenesis induced? If this observation is used to illustrate the early phases of calcium-related cancer research, it is necessary to give readers some information about what was actually observed.

Lane 302 : “intracellular” in the opinion of the Reviewer, “cytosolic” would be more appropriate, as “intracellular” involves also the intracellular storage organelles, in which calcium levels are not obligatorily low.

Lane 310 : “Ca2+concentration “ : Ca2+ concentration

Lane 320 : “Considering that the ion concentration in mammary glands and breast milk is 10 mM and 2-4 mM respectively” : where in the mammary gland? Luminally? In acini? In minor/major ducts? What determines the difference between the 10 mM and the 2-4 mM values?

Lane 337 : “Voltage-dependent ion channels are integral mechanosensitive membrane proteins” : this is somewhat confusing, because in the opinion of the Reviewer, not all voltage-dependent ion channels are obligatorily mechanosensitive.

Lane 371 : “induction of apoptosis [110] and programmed cell death [111]” : please rephrase, as apoptosis is a form of programmed cell death.

Lane 378 and 386 : “Ca2+” : Ca2+

Lane 382 : pleased specify PLC isoform.

Lane 380 : “G-coupled G proteins” : ?

Lanes 386-389 : Please rephrase; for example it is redundant to cite EGF following EGFR, because EGF signals via EGFR. Please include additional PLCg-coupled growth factor receptors (such as PDGFRA etc.) with appropriate References.

Calmodulin is a key intracellular calcium-sensing regulator of cellular signaling. This protein should be absolutely discussed in this work.

Lane 449 : “secretory pathway of calcium ATPase” : as already indicated earlier by the Reviewer, this protein is called “secretory pathway calcium ATPase” (meaning the calcium ATPase of the secretory pathway). Please use name correctly.

Lane 451 : “sorting from the cytosol to Golgi vesicles” : this requires some clarification, because sorting usually occurs between the ER and the Golgi complex. If sorting directly from the cytosol into the Golgi complex is mentioned, corresponding mechanisms need to be briefly mentioned.

Lane 457 : “SPCA2 is frequently overexpressed in patients with hormone receptor-positive and poor prognosis” : meaning in the tumors of the patients?

“where it exerts a pro survival effect on mammary epithelial cells” : on normal or tumoral epithelial cells?

Lanes 464-467 : Please explain the link between SPCA-Orai effects and SPCA effects on redox stress and chemosensitivity to DNA agents, activation of (ATM/ATR), p53 apoptosis promotion.

Table 3. “Altered calcium ion channels in breast cancer. “ : STIM is not an ion channel.

Table 4 : “NFkB” : NF-kB ("kappa"). “Serca” : SERCA

Lane 492 : “Calcium signaling is also linked to mitogen-activated protein kinases (MAPK, MAPK/ERK, Ras-Raf-MEK-ERK), a master kinase involved in extracellular signaling transduction” : Please rephrase, as the mitogen-activated protein kinases are not a (master) kinase, but a cascade of several kinases.

Lane 516 : “intracellular Ca2+ concentration in MCF-7 breast cancer cell line…” : in the MCF-7…

Lane 577 : “but not all agree on the same” : same what?

English needs some editing by a native speaker for clarity.

Author Response

Consulte el archivo adjunto.
